# The Impact of a New Arterial Intravascular Pump on Aorta Hemodynamic Surrounding: A Numerical Study

**DOI:** 10.3390/bioengineering9100547

**Published:** 2022-10-13

**Authors:** Yuan Li, Yifeng Xi, Hongyu Wang, Anqiang Sun, Xiaoyan Deng, Zengsheng Chen, Yubo Fan

**Affiliations:** Key Laboratory of Biomechanics and Mechanobiology (Beihang University), Ministry of Education, Beijing Advanced Innovation Center for Biomedical Engineering, School of Biological Science and Medical Engineering, Beihang University, Beijing 100083, China

**Keywords:** cardiopulmonary syndrome, intravascular pump, renal perfusion, high shear stress, residence time

## Abstract

*Purpose:* The purpose of this study was to investigate the impact of a new arterial intravascular pump on the hemodynamic surroundings within the aorta. *Methods:* A new arterial intravascular pump was placed in the descending aorta, and the effects of three positions within the aorta, as well as the number (n = 1 to 3) of pumps, on arterial flow features, organ perfusion, and blood trauma were investigated using a computational fluid dynamics (CFD) method. *Results:* It was found that as the pump position was moved backward, the perfusion in the three bifurcated vessels of the aorta arch increased and the pump suction flow decreased, resulting in a reduced high shear stress and decreased residence time in the three branches of the aortic arch. The further posterior the location of the pump, the better the blood flow perfusion to the kidneys, while the perfusion at the bifurcation of the abdominal aorta was reduced, due to the pump suction effect. Compared to the condition with single pump support, the multi-pump assist model can significantly reduce the pump rotating speed, while keeping the same flow patterns, leading to a decreased volume of high shear stress and flow loss. When increasing the number of pumps, the perfusion to the three branches of the aortic arch increased, accompanied by a diminished residence time, and the perfusion to the other aortic branches was decreased. However, the perfusion to the other aortic branches, especially for the renal arteries and even under a three-pump condition, was close to that without pump assistance. *Conclusion:* The placement of an intravascular pump near the beginning of the suprarenal abdominal aorta was considered the optimal location, in order to improve the hemodynamic surroundings. Increasing the number of pumps can significantly reduce the rotational speed, while maintaining the same flowrate, with a decreased fluid energy loss and a reduced high shear stress. This arterial intravascular pump can effectively improve renal blood flow.

## 1. Introduction

Cardiorenal syndrome (CRS) is a complex pathophysiological disorder of the heart and the kidneys, in which acute or chronic dysfunction in one organ may lead to acute or chronic abnormality in another [1,2]. Kidney disease has emerged as an independent risk factor for the development of congestive heart failure (HF) and acute ischemic cardiomyopathy events [3,4,5]. The proportion of patients admitted with heart failure who develop acute cardiorenal syndrome is 25% to 40% in the United States and Europe, and 32% to 44% in China [6]. 

A clinical report showed that the in-hospital mortality of CRS was 23.2% [7]. CRS is a complex disease [8,9], which may contribute to acute kidney injury, decompensated heart failure, cardiogenic shock, and cardiac surgery-associated low cardiac output syndrome. How to effectively improve the treatment of patients with cardiorenal syndrome, as well as prolong the life of patients, is an urgent need at present.

The pathophysiology of renal injury due to heart failure is multifactorial, and one of the most important factors is low renal perfusion [10]. In order to improve renal perfusion and assist blood circulation for heart failure patients, a new miniature axial blood pump-assisted method has been proposed. This blood pump is known as an arterial intravascular pump and has a small diameter (about 6 mm) [11,12,13,14]. It can be implanted quickly and is usually placed in the descending aorta, to assist blood perfusion [15]. Clinical studies have demonstrated that intravascular pumps help to increase urine output and improve renal function for cardiorenal syndrome patients [12]. The same conclusion was reached by Shiva et al. [13], who implanted this blood pump in eight pigs and found that it could help to reduce left ventricle afterload, increase renal arterial pressure, and increase cardiac flow output. Recently, Puzzle Medical Devices Inc (Montreal, Quebec, Canada) proposed a modular design for an arterial intravascular pump. Specifically, three arterial intravascular pumps were implanted sequentially through the femoral artery and then assembled together using transcatheter technology, so that the blood pumps can function simultaneously. This design holds promise for safe percutaneous implantations, with a low risk of bleeding, stroke, or pump thrombosis, while providing sustained symptom relief, reducing rehospitalization, and improving overall life quality [16].

Up to now, there has been a lack of information about intravascular pumps; there have only been a few studies focusing on animal models and clinical case reports [11,12,13,14], which were limited by their experimental methods and also could not specifically analyze the effects of the hemodynamic surrounding, organ perfusion, and blood trauma resulting from arterial intravascular pump use, nor could they exhaustively investigate the effects of different pump placements and numbers of pumps. Understanding the impact of this interventional pump on the hemodynamic environment within the aortic arch can better assist clinical treatment and the optimal design of this type of interventional pump, for reducing clinical complications and improving clinical treatment. To determine the mechanisms of impact of interventional pumps on the aorta surroundings, this study was the first to use computational fluid dynamic (CFD) methods [14,17,18,19], using the commercial software ANSYS CFX, to assess the effects of the placement and number of such a pump on intra-aortic blood flow patterns, organ perfusion, and blood damage. Specifically, we obtained a model of the artery of the patient using clinical CT and modified the commercial pump Impella 5.0 to meet the requirements of an arterial intravascular pump. The intravascular pumps were placed at three different positions. The use of one, two, or three intravascular pumps was tested in this study. Flow surrounding was described using energy loss metrics and the flow field, organ perfusion was assessed by calculating the outlet flow percentage, and blood damage was assessed by calculating the shear stress and residence time. As indicated by our study, the position the of intravascular pumps had an important impact on the hemodynamic surroundings of the aorta, and increasing the number of pumps can significantly reduce the pump rotating speed, while keeping the same flowrate, leading to a decreased high shear stress and flow loss, which can help to reduce the pump-induced blood trauma and improve the hemocompability of intravascular pumps.

## 2. Materials and Methods 

### 2.1. Studied Model and Variations

This study was approved by the Institutional Review Board at the School of Biological Science and Medical Engineering at Beihang University. The commercial software Mimics (Materialise, Belgium) was employed to construct a human arterial vascular model from CT data (Figure 1a). This arterial model contains 12 branches, which include the brachiocephalic trunk (outlet 1), left common carotid artery (outlet 2), left subclavian artery (outlet 3), left gastric artery (outlet 4), hepatic artery (outlet 5), splenic artery (outlet 6), superior mesenteric artery (outlet 7), right renal artery (outlet 8), left renal artery (outlet 9), inferior mesenteric artery (outlet 10), right iliac artery (outlet 11), and left iliac artery (outlet 12) (Figure 1a). The arterial intravascular pump used in this study was modified by Impella 5.0 (Abiomed Inc., Danvers, MA, USA) and included a rotor placed in the casing, a diffuser, and a motor (Figure 1b). This study investigated the influence of arterial intravascular pumps in cardiopulmonary syndromes and the impact of the position and number of intravascular pumps located in the descending aorta on organ perfusion, arterial flow features, and blood trauma. (Figure 1c). The intravascular pumps were placed at the beginning of the descending thoracic aorta (Location 1), at the beginning of suprarenal abdominal aorta (Location 2), and at the end of the suprarenal abdominal aorta (Location 3) [13] (Figure 1c). The use of one, two, or three intravascular pumps was tested in this study (Figure 1c). 

### 2.2. CFD Methods

In this study, the blood had a density of 1055 kg/m^3^ and a viscosity of 0.0035 Pa·s [18]. Reynolds-averaged Navier–Stokes (RANS) equations were solved using the commercial software Ansys CFX (ANSYS Inc., Canonsburg, PA, USA), which employs a finite-volume method, based on discretization of governing equations. The convective terms were solved in high-resolution form, and the SST k-ω turbulence model [18,20] was employed for stabilization simulations [21,22]. In addition, a normal artery without an intravascular pump (no-pump case) was analyzed, to calculate the normal/base levels, as the control group for evaluating the effect of adding an intravascular pump, in which the blood was considered as laminar flow. The aortic root was set as the inlet boundary condition, and a mass flow rate of 0.079 kg/s was set (obtained from a volume flow rate of 4.5 L/min), which was generated by intravascular pumps. The end surface of each arterial branch (numbered 1–12) was set as the outlet boundary condition, and the end surface pressures were obtained from clinical data (Table 1). The frozen-rotor interface was placed near the rotor domain, and the interface was set to rotate, to introduce rotational effects, while the rest of the domain (the aorta, inlet pipe, and diffuser of the arterial intravascular pump and so on) was stationary. All vessel walls were assumed to be no-slip and adiabatic, and the convergence criterion was set to 10^−6^. The intravascular pump rotating speed was set according to Table 2, so that it met the given flow rate.

### 2.3. Scalar Shear Stress Predictions

The viscous scalar shear stress (SSS) was derived from the simulated flow field, according to the following equation [17,24]:(1)σ=16∑σii−σjj2+∑σijσij12
where σ is the shear stress tensor, which was calculated by multiplying the shear rate tensor $\;\sigma_{\mathrm{ij}}=\partial v_{i}/\partial x$ with the blood viscosity [25].

### 2.4. Residence Time Predictions 

Residence time (RT) was applied to determine the areas of stagnation and flow re-circulation. The deposition of blood components (platelets, fibrinogen, and so on) in regions with a long residence time can lead to the formation of thrombosis [26,27]. The Eulerian residence time obeys the following equation [28]:(2)$$\frac{\partial RT}{\partial t}+v\cdot \nabla RT=D_{RT}\nabla^{2}RT+1$$
where t is time, v is the velocity of blood, and *D_RT_* = 1.14 × 10^−11^ m^2^/s [28] represents the self-diffusivity of blood, and the source is consistent with the variation of time. 

In the commercial software ANSYS CFX, the residence time calculation is performed by rewriting the transport equations. Specifically, by creating subdomains and calling the transport equations and defining source terms in the subdomains, the blood damage metric can be solved in parallel with the underlying physical quantities (such as the pressure, velocity, and so on).

### 2.5. Energy Loss Predictions 

To identify the effect of the intravascular pump on the arterial blood flow, the concept of energy losses was introduced in this study and evaluated by means of a dissipation function, defined as in [29,30]:(3)$$\emptyset =\frac{\mu }{2}{\left(\frac{\partial v_{i}}{\partial x_{j}}+\frac{\partial v_{j}}{\partial x_{i}}\right)}^{2}-\frac{2}{3}\mu {\left(\frac{\partial v_{i}}{\partial x_{i}}\right)}^{2}$$
where μ is the amount of eddy viscosity and dynamic viscosity, and $\frac{\partial v}{\partial x}\;$is the velocity gradient.

To show the energy losses of the artery clearly, the dissipation function was considered to be logarithmic:log_10_(*Φ*) = log_10_(dissipation function)(4)

When the value of log_10_ (dissipation function) is high or low, this means the magnitude of energy losses is long or small, respectively.

### 2.6. Mesh Details and Sensitivity Analysis

Ansys ICEM (ANSYS Inc., Canonsburg, PA, USA) was employed to generate a tetrahedral mesh with refinement of some geometric features, such as the curvature, outlet diameter, and branching. To capture the flow near the walls, five-layer prismatic meshes were employed as boundary layers. Their y pulse was approximately equal to 1, which satisfies the requirements for the solution of the SST k-ω turbulence model. All meshes in this study had skew angles greater than 30°, aspect ratios less than 1.5, and were without a negative volume. To ensure the accuracy of the calculation results, on the one hand, the inlet and outlet of the model were sufficiently extended (about 50-times the diameter), to avoid the influence of boundary conditions on the simulation results. On the other hand, grid-independent verification was employed, and important indexes of the study, such as each organ’s perfusion, shear stress, and so on, were observed as the mesh number increase or decrease; finally, a mesh number of 9 million (Figure 2) was chosen for the subsequent study, based on the consideration of simulation accuracy and simulation time.

## 3. Results 

### 3.1. Effect of Different Positions of the Arterial Intravascular Pump

The rotating speed of the intravascular pump in the three different positions was set to 33,000 rpm (Table 2). When the intravascular blood pump started, blood flowed out of the left ventricle, and part of the blood moved into the arterial branch in front of the pump, while the remainder moved through into the pump (Figure 3a). Due to the acceleration of blood by the pump, the flow field after the pump was disturbed (Figure 3a). These disturbances induced energy losses within the artery, which were found to be greatest near the pump outlet and extended downward, in the direction of the blood flow (Figure 3b). It was found that a portion of the blood that had just been pumped was reabsorbed into the blood pump, which could cause secondary damage to the blood behavior (Figure 3a). As the pump position was moved back, the flow rate into the pump decreased (Figure 3c). Since all three positions of the intravascular pump were located after the aortic arch, the suction of the intravascular pump affected the flow rate of the three bifurcated vessels of the aorta arch. The further back the intravascular pump was positioned from the aortic arch, the weaker the effect of the pump on the flow in the aortic arch and its three branches; and thus the greater the blood flow into the three branches of the aortic arch (outlet 1, 2 and 3), the less the blood flow into the pump. Therefore, the flow rate into the pump was reduced (the suction power of the pump decreased) as the position was shifted backward (Figure 3c). The percentage of the outlet flow rate of the three branches in the aortic arch increased as the intravascular pump position was moved backward within the aorta (10.2% in Location 1, 16.3% in Location 2, and 18.1% in Location 3) (Figure 3d). It is worth pointing out that, when the pump was located at the beginning of the descending thoracic aorta (Location 1) and at the beginning of suprarenal abdominal aorta (Location 2), the flow into the pump was slightly higher than the flow into the aortic arch root, which can be attributed to the secondary suction of blood around the pump (Figure 3a). Compared to the no-pump case, the percentage of outlet flow was reduced after intravascular pump implantation. However, the more posterior the position of the intravascular pump, the better the perfusion to both kidneys (outlet 8 and 9) (28.7% in Location 1, 28.8% in Location 2, and 29.3% in Location 3) (Figure 3d). It was noteworthy that, when the intravascular pump was located at the end of the suprarenal abdominal aorta (Location 3), its strong suction action led to a very low flow of the gastric artery (outlet 4), hepatic artery (outlet 5), splenic artery (outlet 6), and superior mesenteric artery (outlet 7) (Figure 3d).

A high rotating speed of the intravascular pump generated an extremely high non-physiological shear stress on the arterial vessel posterior to the rotor (Figure 4a). As the position of the intravascular pump was moved downstream, less blood was accelerated by the pump (Figure 3c), and there was less impingement on the arterial vessel and less energy losses (Figure 3b); thus, the high shear stress on the arterial vessel created by the blood pump was reduced (Figure 4a), which is similar to the distribution of energy losses region within the artery (Figure 3b). As shown in Figure 4b, all the shear stress volumes at a level of larger than 50 Pa, 100 Pa, and 150 Pa decreased with the posterior movement of the intravascular pump position. According to previous studies [13,31,32,33], shear stress volumes greater than 50 Pa are highly correlated with platelet activation, and shear stress volumes greater than 150 Pa are strongly correlated with red blood cell damage. In the present study, high shear stress-caused blood damage decreased with the posterior movement of the intravascular pump. Regarding the residence time, as the intravascular pump position was moved backward, the blood perfusion in the three branches of the aortic arch (outlet 1, 2, and 3) increased (Figure 3d), thus the residence time of blood in these branches decreased (Figure 4c), which indicates that the blood stagnation and thrombotic potential also decreased.

### 3.2. Effects of Different Numbers of Arterial Intravascular Pumps

A multiple intravascular pump model has recently been proposed [16]. Compared with single pump support, multiple pump support reduces the blood velocity in and out of the intravascular pump (Figure 3a and Figure 5a), which also results in lower energy losses in the arteries (Figure 3b and Figure 5b). Multi-pump support significantly reduces the rotating speed, while keeping the same flow rate, compared to a single pump support (Figure 5c). The greater the number of pumps employed, the lower the pump speed needed to create the same flow rate (33,000 rpm for single-pump support, 17,500 rpm for two-pump support, and 12,000 rpm for three-pump support). A multi-pump assistance model also has a significant circulating blood flow that re-enters the blood pump, leading to secondary damage to the blood components (Figure 5a). When two intravascular pumps are employed for support, the reduced pump rotating speed results in a lower pump suction to the three bifurcated vessels of the aorta arch (outlet 1, 2, and 3), allowing their perfusion to increase, compared to single pump support (Figure 5d). The blood flow supplied to the kidneys (outlet 8 and 9) and lower extremities (outlet 11 and 12) for the two intravascular pump condition was lower than that of the single pump condition, while being almost the same as the no-pump (base) case (Figure 5d). With three intravascular pumps, the perfusion in the three bifurcated vessels of the aorta arch (outlet 1, 2, and 3) was high, due to a further reduction of pump suction (Figure 5d). For the three intravascular pump condition, the blood flow supplied to the kidneys (outlet 8 and 9) and the other abdominal organs was similar to that of the two intravascular pump or without pump conditions, while the perfusion to the lower extremities (outlet 11 and 12) was slightly lower.

The multi-pump assist model reduced the blood velocity (Figure 5a) close to the pump and decreased the energy losses (Figure 5b), due to the significantly lower rotating speed. The non-physiological shear stress generated at the arterial vessel decreased as the number of intravascular pumps increased (Figure 6a). All the volumes with a high level of shear stress larger than 50 Pa, 100 Pa, and 150 Pa were reduced with the increasing number of intravascular pumps (Figure 6c). The multiple pumps (two or three pumps) changed the volume of the high shear stress larger than 100 Pa and 150 Pa to a very small level, or it even disappeared (Figure 6c). This indicates that as the number of pumps increased, the shear stress-caused damage to the arterial vessel and blood components reduced and the hemocompatibility improved. In the case of multiple pump assistance, the blood perfusion in the three branches of the aortic arch increased with the number of pumps (Figure 5d). The residence time of blood in the three branches of the aortic arch was reduced (Figure 6c), and the thrombotic potential due to blood stagnation decreased.

## 4. Discussion

Heart failure is a pathophysiological and complex disease, with an extremely high mortality and rehospitalization rate [34]. Hemodynamic disturbances (such as variations in vascular resistance, impaired cardiac output, and fluid overload) due to heart failure can lead to inadequate renal perfusion and also induce cardiorenal syndrome, with significant effects on both glomerular filtration and tubular reabsorption [35]. As an extension of ventricular assistance devices, a new concept of arterial intravascular pump was proposed by Puzzle Medical Devices Inc. [16] that can be installed in the descending aorta to provide blood flow support to the kidneys of patients with cardiorenal syndrome. This study is the first numerical simulation analyzing the flow behavior of the new arterial intravascular pump on the hemodynamic surroundings within the aorta. The effects of the location and number of intravascular pumps within the aorta on the flow features, organ perfusion, and blood damage were evaluated. As shown by the study, when the intravascular pump was located at the beginning of the descending thoracic aorta (Location 1), the strong suction resulted in less perfusion in the three bifurcated vessels of the aorta arch, which could cause insufficient blood supply to the brain and upper extremities. In addition, this was accompanied by a high residence time in the aorta arch branches, leading to a high risk of blood stagnation, and even thrombosis. Additionally, when the intravascular pump was placed at Location 1, the blood flow moving into the intravascular pump was high and the energy losses was large, and the volume of high level shear stress was high, which could induce severe blood damage. When the pump was placed at the end of the suprarenal abdominal aorta (Location 3), although the perfusion to the three bifurcated vessels of aorta arch was increased and the volume of high shear stress was low, its powerful suction greatly reduced the blood perfusion to the branch arteries in the abdomen aorta (gastric artery, hepatic artery, splenic artery, and superior mesenteric artery), which could cause related complications and worsen the patient’s condition. In comparison with Location 1 and Location 3, Location 2 (the beginning of the suprarenal abdominal aorta) is the optimal position for improving the hemodynamic surroundings. On the one hand, Location 2 increases perfusion to the three bifurcated vessels of the aorta arch compared to Location 1, and Location 2 also does not suck back the blood flow perfused to the branch arteries in abdomen aorta, unlike Location 3, avoiding potential related complications, such as strokes and abdominal organ injuries. On the other hand, Location 2 results in relatively small energy losses and a low volume of high shear stress, indicating low blood damage. The results of this study suggest that when the interventional pump placement is performed, the position affects several of the aorta hemodynamic surrounding factors, and special care needs to be given to a proper placement, to reduce related complications and improve patient recovery. 

This article also examined the impact of increasing the number of blood pumps on the aorta hemodynamic surrounding, and few studies of this type have been performed before. As indicated by the study, an increase in the number of intravascular pumps can significantly reduce the rotating speed of each pump, while maintaining the same circulated blood flow. Compared to the support with a single pump, the rotating speed of each pump is reduced by almost half with the support of two pumps (33,000 rpm vs. 12,000 rpm) and the rotating speed is reduced by almost two-thirds with the support of three pumps (33,000 rpm vs. 12,000 rpm), indicating less blood damage. When one intravascular pump assists independently, the high rotating speeds create a large suction, which not only leads to less perfusion in the three bifurcated vessels of the aorta arch compared to the multi-pump situation and the no-pump (base) condition, but also generates high energy losses and a high volume of high shear stress. When multiple intravascular pumps assist together, not only is the blood flow percentage in each bifurcated vessel of the artery close to the no-pump (base) condition, but there are also reduced energy losses, especially a low volume of high shear stress. It is hard to create a percentage of high shear stress larger than 150 Pa with two pump support, and 100 Pa with three pumps support, and thus the potential blood damage is greatly reduced. Another benefit of using multiple pumps is that the blood flow produced by each pump can be summed, and the flow of each pump reduced; thus, a smaller diameter blood pump can be used, which allows a faster intervention and removal of intravascular pumps with less trauma to the patient. Additionally, multiple blood pumps also provide multiple levels of protection. In the case of failure of one pump during operation, the other pumps can continue to work to maintain blood circulation, and the faulty pump can be removed and replaced quickly. 

It is worth mentioning that in the present study, regardless of the position and the number of the intravascular pumps, there was an increase in perfusion to the renal artery compared to the normal condition without a blood pump. Clinical and animal studies have shown that the use of arterial intravascular pumps significantly improves key hemodynamic parameters in patients and prompts the kidneys to remove more than 10 liters of excess fluid, greatly improving creatinine levels (a measure of kidney function) [12,13]. Furthermore, clinical studies have also found that arterial intravascular pumps can reduce the left ventricular afterload and increase the cardiac flow output [12]. 

This study also considered a dissipation function, to evaluate the energy losses associated with intravascular pumps installed in the descending aorta. It was found that the variation of energy losses in the vessel was similar to the distribution of non-physiological shear stress (Figure 3b and Figure 4a, Figure 5b and Figure 6a). A further correlation investigation showed that the energy losses were well correlated with non-physiological shear stress (Table 3), which means that the blood pump-caused energy losses were related to blood trauma. This highlights the advantages of the multi-pump assist model (modular design), which effectively reduces energy losses, while maintaining perfusion for the branches of the aorta.

This study also has some limitations. First, the blood within the whole artery was considered as a Newtonian fluid in a turbulent state. Under the influence of the intravascular pump, most of the blood within the artery behaved as a Newtonian fluid in the turbulent state, but some of the blood showed the features of a laminar non-Newtonian fluid. Although some studies have shown that this assumption has little impact on the conclusions [15,36], it may still lead to some deviations. Second, because this new blood pump has not yet been widely used in clinical treatment, real clinical data on changes in cardiac output flow after pump implantation are lacking. In the present study, a constant boundary condition was utilized at the inlet and at the outlet of the artery. Finally, although the clinical results support the conclusions of this study, further experimental validation is needed, for a more detailed and accurate mechanistic exploration. For example, the flow rate of each of the arterial branches, as well as the flow rate of the pump suction and damage to blood cells, were measured experimentally when the location or number of the blood pumps were changed.

## 5. Conclusions

During the placement of an arterial intravascular pump, the location may affect the flow features and perfusion of bifurcated vessels within the aorta, and this must be considered. In the present study, placing the intravascular pump near to the beginning of suprarenal abdominal aorta (Location 2) was considered the best position, with improved hemodynamic surroundings. Increasing the number of intravascular pumps can significantly reduce the rotational speed, while keeping the same flow, which can improve organ perfusion, decrease fluid energy loss, and reduce the high shear stress and residence time. The placement of an arterial intravascular pump can effectively improve renal perfusion compared to the condition without a pump. This study can be used to guide the clinical treatment and optimization of blood pump surgical interventions.

## Figures and Tables

**Figure 1 bioengineering-09-00547-f001:**
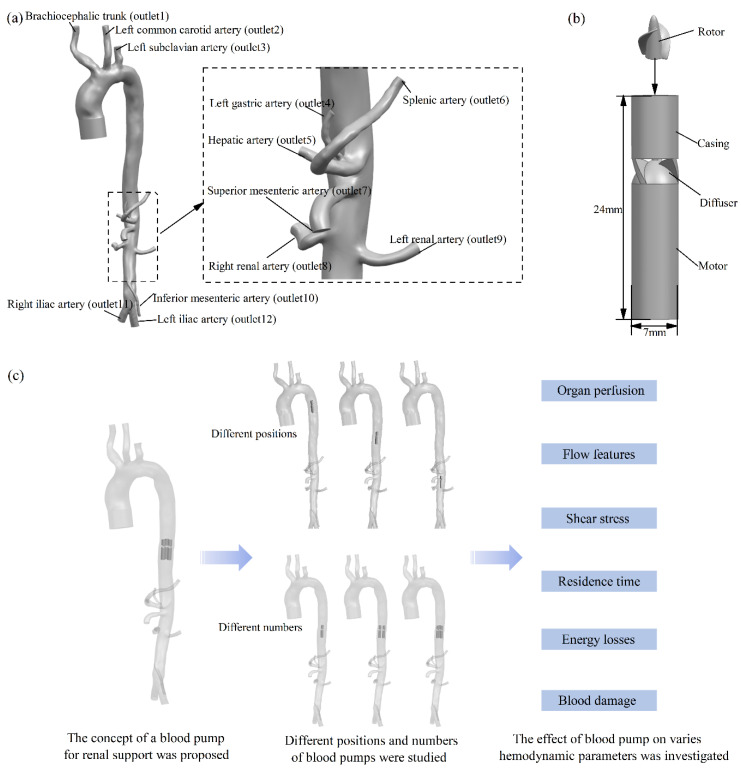
Studied models: (**a**) human arterial model; (**b**) modified Impella 5.0 blood pump for intra-arterial fluid behavior; (**c**) the three different positions and quantities of arterial intravascular pumps studied.

**Figure 2 bioengineering-09-00547-f002:**
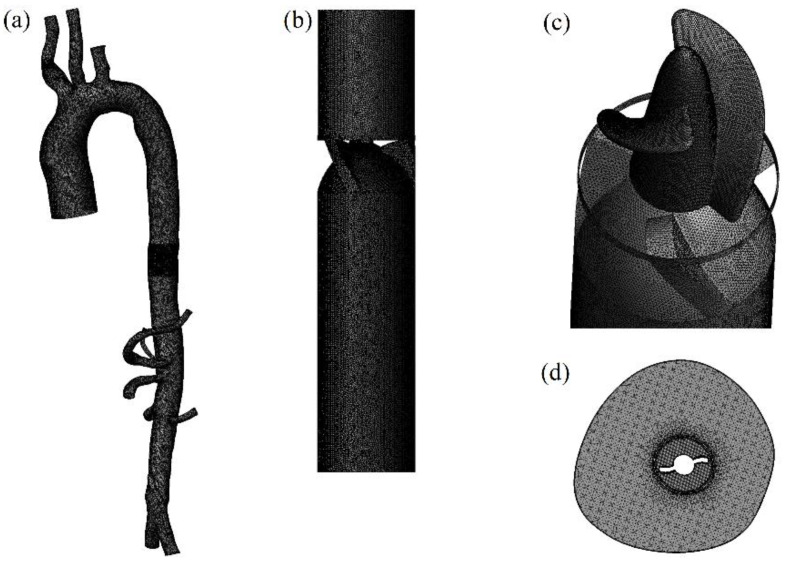
Arterial and studied pump meshes for computational fluid dynamics: (**a**) arterial mesh; (**b**) studied pump mesh; (**c**) studied pump rotor and diffuser mesh; (**d**) cross-sectional mesh of the region where the studied pump was located.

**Figure 3 bioengineering-09-00547-f003:**
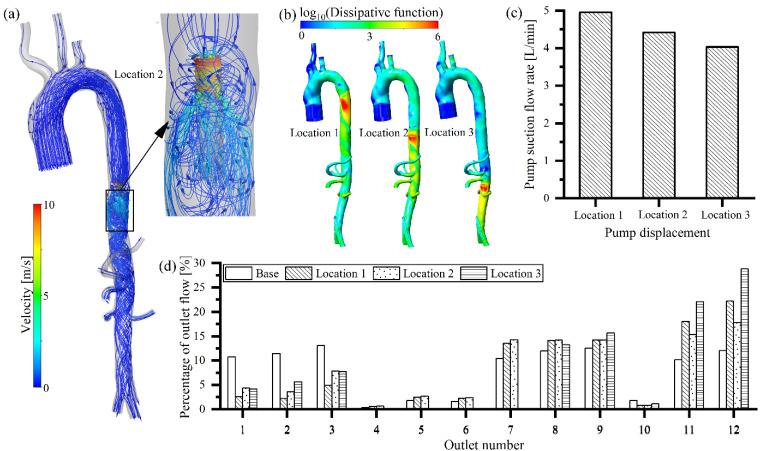
Effects of the different locations of the arterial intravascular pump on flow features: (**a**) effect on arterial flow field; (**b**) effect on pump suction flow; (**c**) effect on energy losses, and (**d**) effect on organ perfusion.

**Figure 4 bioengineering-09-00547-f004:**
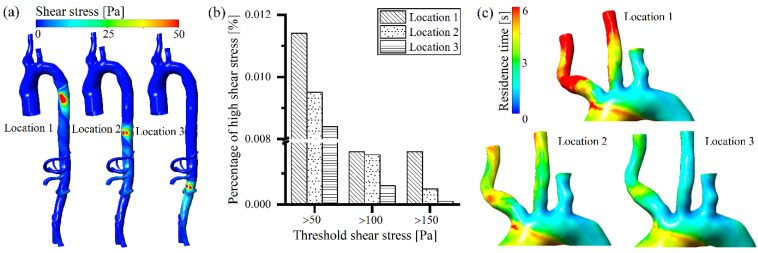
Effects of different pump positions on blood damage-related parameters (shear stress and residence time): (**a**) arterial shear stress distribution; (**b**) high shear stress volume comparison; (**c**) residence time distribution on aorta arch.

**Figure 5 bioengineering-09-00547-f005:**
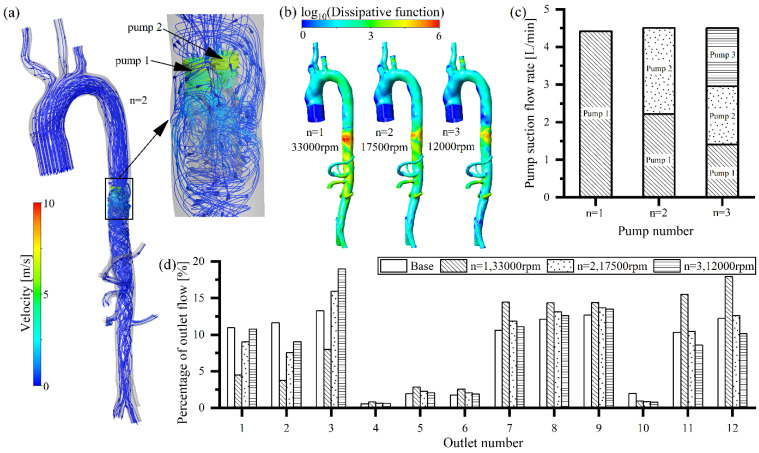
Effects of different numbers of arterial intravascular pumps on the flow features: (**a**) effect on arterial flow field; (**b**) effect on pump suction flow; (**c**) effect on energy losses, and (**d**) effect on organ perfusion.

**Figure 6 bioengineering-09-00547-f006:**
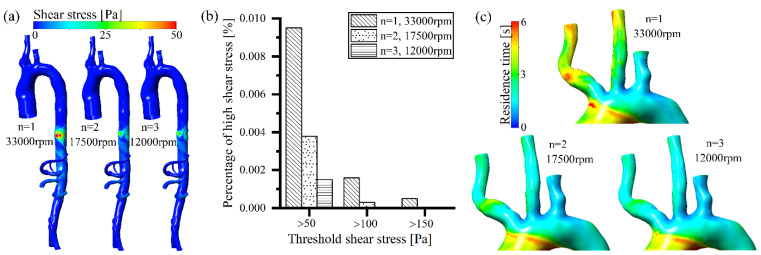
Effects of different pump positions on the blood damage-related parameters (shear stress and resident time): (**a**) effect on arterial shear stress distribution; (**b**) effect on high shear stress volume; (**c**) effect on residence time distribution.

**Table 1 bioengineering-09-00547-t001:** Mean pressure of the arterial branch end surface, based on clinical measurements.

Outlet Number	1	2	3	4	5	6	7	8	9	10	11	12
Static Pressure (mmHg)	101	100	102	103	103	103	103	102	99	103	104	104

**Table 2 bioengineering-09-00547-t002:** The relationship between the rotating speed and flow rate of Impella 5.0 [23].

Rotating Speed (rpm)	Flow Rate (L/min)
10,000	0.0–1.4
17,000	0.5–2.6
20,000	0.5–3.1
22,000	0.9–3.4
24,000	1.4–3.7
26,000	1.8–4.0
28,000	2.6–4.4
30,000	3.4–4.7
33,000	4.2–5.3

**Table 3 bioengineering-09-00547-t003:** Spearman’s rank correlation analysis of mean energy losses and non-physiological shear stress level.

Energy Losses	Variable	r^2^
Mean dissipation	Shear stress above 50 Pa	0.96
	Shear stress above 150 Pa	0.72
	Shear stress above 100 Pa	0.68
	Shear stress above 50 Pa	0.78

## Data Availability

Not applicable.

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
