# Peer review of "The Impact of a New Arterial Intravascular Pump on Aorta Hemodynamic Surrounding: A Numerical Study"

_bioengineering, 2022, doi:10.3390/bioengineering9100547_

Round 1

Reviewer 1 Report

Review of the manuscript entitled:The impact of a new arterial intravascular pump on aorta hemodynamic surrounding: a numerical study.

In this work, the effect of the position of a micropump placed in the aortic artery is investigated.

The work is very interesting and well written, I think it can be accepted in the magazine, and I have only a couple of minor comments to correct.

1. You must clarify if the mesh where the pump is rotated or fixed, if it is broken, you must explain the condition imposed between the two meshes.

2. A fixed aortic artery inlet velocity was used, but the velocity changes during a heartbeat, explain how this affects the results obtained.

3. You must explain better why the suction power of the pump decreases when you change its position.

4. How did you solve the residence time equation?

5. How does this study compare with those that use only the pulse of the pump but do not include the pump in the artery, because you place the pump in the artery, that is how it is done clinically?

Author Response

We would like to take this opportunity to thank the reviewers for their thoughtful and constructive reviews of our manuscript and their appreciation of our work. The comments and suggestions of the reviewers have helped us improve the manuscript. In the revised manuscript, the revised texts were highlighted in red to address the comments of Reviewer 1. The below are our point-to-point responses to the comments and suggestions by the reviewers. The original review comments are in italic format and the corresponding response follows each comment.

Reviewer 2 Report

The present manuscript shows that the thoracic aorta blood flow is directly affected by the number of pumps located in some strategic parts of the artery producing different behaviors in the flow patterns. Numerical simulations were performed through the finite element method (CFD). The study's outcomes highlight important flow patterns into the artery model according to the number of blood pumps and the rpm working function of each pump. These effects require constant medical attention. This manuscript is fascinating in the biomechanics field. However, in my opinion, it cannot be published without significant corrections. Some concerns must be clarified, and the manuscript must be better written.

In the last paragraph of the introduction, the authors should highlight the way the present study is novel regarding other of the same topic in the literature. The novelty is not well emphasized.

A flowchart must be considered to show step-by-step the procedure and solution of the present study.

Some critical information is missing. For example, which type of mesh are you considering and why? Or how do you know that the current mesh is accurate to obtain precise results?

 Which phase are you considering in the CFD solution? Is it Diastole or Systole?

There are grammar and style errors throughout the paper. In addition, the writing needs substantial improvement (grammar and spelling)

The aim of the present work, numerical simulation of the behavior flow patterns according to the number of pumps and non-invasive methods, requires more explanations. Also a clinical perspective of this research is needed. 

Author Response

We would like to take this opportunity to thank the reviewers for their thoughtful and constructive reviews of our manuscript and their appreciation of our work. The comments and suggestions of the reviewers have helped us improve the manuscript. In the revised manuscript, the revised texts were highlighted in red to address the comments of Reviewer 2. The below are our point-to-point responses to the comments and suggestions by the reviewers. The original review comments are in italic format and the corresponding response follows each comment.

Round 2

Reviewer 2 Report

The authors have enhanced the work, and I agree with the explanation and improvements.

I would only request a final review of the typo and modify the appearance of figure 2

Congratulations on the idea and the result